

# Insights on ozone pollution control in urban areas by decoupling meteorological factors based on machine learning

Yuqing Qiu[1], Xin Li[1,2*], Wenxuan Chai[3*], Yi Liu[1], Mengdi Song[1], Xudong Tian[4], Qiaoli Zou[4], Wenjun Lou[5], Wangyao Zhang[5], Juan Li[5] and Yuanhang Zhang[1]

[1]College of Environmental Sciences and Engineering, Peking University, Beijing 100871, China

[2]Institute of Carbon Neutrality, Peking University, Beijing 100871, China

[3]China National Environmental Monitoring Center, Beijing 100012, China

[4]Zhejiang Ecological and Environmental Monitoring Center, Hangzhou 310012, China

[5]Jinhua Ecological and Environmental Monitoring Center, Jinhua 321015, China

*Correspondence to: Xin Li (li_xin@pku.edu.cn),Wenxuan Chai(chaiwx@cnemc.cn)*

**Abstract.** Ozone ($O_3$) pollution is posing significant challenges to urban air quality improvement in China. The formation of surface $O_3$ is intricately linked to chemical reactions which are influenced by both meteorological conditions and local emissions of precursors (i.e., NOx and VOCs). The atmospheric environment capacity decreases when meteorological conditions deteriorate, resulting in the accumulation of air pollutants. Although a series of emission reduction measures have been implemented in urban areas, the effectiveness of $O_3$ pollution control proves inadequate. Primarily due to adverse changes in meteorological conditions, the effects of emission reduction are masked. In this study, we integrated machine learning model, the observation-based model and the positive matrix factorization model based on four years of continuous observation data from a typical urban site. We found that transport and dispersion impact the distribution of $O_3$ concentration. During the warm season, positive contributions of dispersion and transport to $O_3$ concentration ranged from 12.9% to 24.0%. After meteorological normalization, the sensitivity of $O_3$ formation and the source apportionment of VOCs changed. The sensitivity of $O_3$ formation changed from the NOx-limited regime to the transition regime between VOC- and NOx-limited regimes during the $O_3$ pollution event. Vehicle exhaust became the primary source of VOC emissions after removing the effect of dispersion, contributing 41.8% to VOCs during the pollution periods. On the contrary, the contribution of combustion to VOCs decreased from 33.7% to 25.1%. Our results provided new recommendations and insights for implementing $O_3$ pollution control measures and evaluating the effectiveness of emission reduction in urban areas.



## 1 Introduction

Ozone ($O_3$) plays a significant role in atmospheric oxidation and global climate. It is also considered one of the major atmospheric pollutants. High concentration of surface $O_3$ is harmful to human health, such as causing respiratory diseases and even cancer (Cohen et al., 2017; Monks et al., 2015). In recent

years, China has been in a stage of rapid economic development, accompanied by the emergence of various air pollution problems due to industrialization and urbanization. (Zhang et al., 2012). In order to deal with the air pollution, the Chinese government has issued some control policies, such as Clean Air Action Plan in 2013 (Chinese State Council, 2013) and Blue-Sky Protection Campaign in 2018 (Chinese State Council, 2018). These policies have resulted in reductions in the concentrations of

particulate matter (PM), nitrogen dioxide ($NO_2$) and sulfur dioxide ($SO_2$) (Zheng et al., 2018). On the contrary, $O_3$ pollution has become increasingly serious, especially in the typical urban clusters such as the Beijing-Tianjin-Hebei (BTH), the Yangtze River Delta (YRD) and the Fenwei Plain (FWP). In 2022, the 90th percentile of maximum daily 8 h average (MDA8) $O_3$ were 179 $\mu g/m^3$ in the BTH, 162 $\mu g/m^3$ in the YRD and 167 $\mu g/m^3$ in the FWP, 4.7%, 7.3% and 1.2% higher than that in 2021,

respectively (Ministry of Ecology and Environment of China, https://www.mee.gov.cn/). Frequent $O_3$ pollution events have attracted the attention of the public and the government. Surface $O_3$ is mainly formed by the photochemical reactions of volatile organic compounds (VOCs) and nitrogen oxides ($NOx = NO + NO_2$) (Atkinson, 2000). The emissions of precursors effectively affect the change of $O_3$ concentration (Tan et al., 2018). The sources of VOCs are complex and widespread, making it

challenging to control emissions. Meteorological conditions can directly or indirectly affect $O_3$ concentration (Liu and Wang, 2020; Zhang et al., 2015). Wind and boundary layer height influence the diffusion of the concentrations of $O_3$ and its precursor. Poor dispersion can result in a decrease in atmospheric environmental capacity, making $O_3$ pollution events more likely to occur even with low precursor emissions. High ultraviolet radiation and temperature promote photochemical reactions of $O_3$

formation (Yang et al., 2019). In addition, $O_3$ can be transported over long distances due to its the long atmospheric lifetime, which can cause regional $O_3$ problems (Han et al., 2019). In short, the $O_3$ concentration is nonlinear affected by meteorological conditions, emissions of precursors and chemical reactions (Fu et al., 2019; Hu et al., 2021).



Li et al. (2020) discovered that approximately 1/3 of the growth of $O_3$ concentration in summer in China was attributed to meteorological conditions. This indicated that the reduction of air pollutants concentrations due to the control policies may be offset by the deterioration of meteorological conditions. Therefore, decoupling meteorological factors from temporal concentrations series of atmospheric pollutants is helpful to assess the impact of clean air action. At present, many mathematical statistical methods have been developed to remove the influences of meteorological

factors. The technique for predicting air pollutants concentrations under randomly selected meteorological parameters was first introduced by Grange et al. (2018). Weng et al. (2022) found that the temperature near the surface 2 m, the downward radiation flux of the surface and the relative humidity were the most important meteorological factors to affect $O_3$ concentration in China by applying two machine learning algorithms (ridge regression and random forest regression).

Mousavinezhad et al. (2021) employed the Kolmogorov-Zurbenko (KZ) filter method and found that meteorological factors played the dominant role on $O_3$ formation in four typical urban agglomerations in China. Guo et al. (2022) used the random forest method to obtain the characteristics of air pollution in 12 megacities in China from 2013 to 2020, and carried out a comprehensive assessment of the actual impact of the national clean air action. Compared to traditional statistical methods, machine learning

models perform better in removing meteorological effects from concentration data.

In response to severe $O_3$ pollution, a series of emission reduction measures targeting $O_3$ precursors have been implemented in urban areas. However, the effectiveness of controlling $O_3$ pollution fell short of expectations. According to previous studies, $O_3$ formation in urban areas was more sensitive to VOCs (Feng et al., 2019), with anthropogenic emissions of VOCs playing a dominant role (Ahmad et al.,

2017). Understanding the sensitivity of $O_3$ formation and the source characteristics of VOCs are helpful to design effective strategies to control $O_3$ pollution. The basis for precursor emission reduction policies relies on the observation-based model (OBM) or the positive matrix factorization model (PMF), but the model results based on observed data are influenced by fluctuations of meteorological conditions. Wu et al. (2023) developed initial concentration dispersion normalized PMF (ICDN-PMF)

to reflect changes in source emissions of VOCs in Qingdao. The results proved that the contribution of solvent use overestimated due to air dispersion during $O_3$ pollution. Additionally, the actual effectiveness of emission reduction measures can also be obscured by unfavorable meteorological conditions. In this study, we applied the Random Forest (RF) method proposed by Grange et al. (2018)




to remove the dispersion and transport effects on $O_3$ concentration, as well as the dispersion effect on precursors in Hangzhou from 2019 to 2022. After meteorological normalization, the concentrations of VOCs were imported into OBM and PMF to obtain the sensitivity of $O_3$ formation and the contributions of emission sources, providing more accurate results. The interplay of meteorological and local factors on $O_3$ pollution can be evaluated effectively and comprehensively in this method. Our results emphasized the importance of decoupling the meteorological effects of transport and dispersion for understanding the mechanisms of local $O_3$ formation and devising appropriate emission reduction measures.

## 2 Methods

### 2.1 Observation data

The online hourly observation data from 2019 to 2022 were measured by the Zhejiang Ecological and Environmental Monitoring Center (30.29°N, 120.13°E). This station was located in the urban area of Hangzhou, Zhejiang Province, surrounded by residential and commercial areas. The data set of air pollutants included $O_3$, $NO_2$ and 98 different kinds of VOCs detected by gas chromatography system, including 29 alkanes, 11 alkenes, 1 alkyne, 16 aromatics, 28 halohydrocarbons, 12 oxygenated VOCs (OVOCs), and 1 acetonitrile. The online gas chromatography system was equipped with a mass spectrometer and flame ionization detector (GC-MS/FID), which used a dual gas path separation method. VOCs compounds with low carbon numbers ($C_2$-$C_5$) were measured by FID, while VOCs compounds with high carbon numbers ($C_5$-$C_{10}$) were detected by MS. Meteorological parameter contained temperature (T), relative humidity (RH), atmospheric pressure (P), wind speed (WS) and wind direction (WD). In addition, we used the meteorological data from the ERA5 reanalysis product (Hersbach et al., 2020), such as boundary layer height (BLH) and ultraviolet radiation b (UVB). The EAR5 meteorological data is spatial grid data with a resolution of 0.25°×0.25° and available at https://cds.climate.copernicus.eu/cdsapp. The back trajectories were calculated backwards in time for 24 h and started 500 m above ground level by using the Hybrid Single Particle Lagrangian Integrated Trajectory (HYSPLIT) model (Stein et al., 2015). The meteorological data from the Global Data Assimilation System (GDAS) with a horizontal resolution of 1° longitude × 1° latitude were adopted in



trajectory model. The back trajectories were then clustered into five clusters by using the Euclidian distance.

**2.2 Meteorological normalization method**

Random Forest is a versatile classifier that comprises multiple decision trees, applicable to classification, regression, and dimension reduction problems. When constructing each tree in the RF model, a dataset of the same size is selected for training, potentially containing duplicates. This sampling method, which involves putting instances back into the dataset, is referred to as bootstrap. At each node, the optimal segmentation is calculated by randomly selecting a subset of features from the entire set. The RF model describe the relationship between the time series of atmospheric pollutants concentrations and their corresponding feature. We constructed random forest model based on original datasets, which contained air pollutants variables ($O_3$, $NO_2$, total non-methane hydrocarbon compounds (NMHCs) and 98 VOC species), time variables (trend, hour, weekday, month and day of year) and meteorological variables (T, RH, P, WS, WD, UVB, BLH and cluster). T, RH and UVB can characterize the local production and loss by chemical reactions. WD, WS and BLH are crucial for the dispersion of $O_3$ and its precursors on a local scale. While cluster can reflect the effect of transport from remote regions. The parameter 'trend' can indicate the long-term changes of air pollutants concentrations resulting from the implementation of policy measures (Vu et al., 2019), which was calculated as Eq. (1):

$$\text{trend} = \text{year}_i + \frac{t_{JD}-1}{N_i} + \frac{t_H}{24N_i} \tag{1}$$

Where $N_i$ is the number of days in the $\text{year}_i$ ($\text{year}_i$ is from 2019 to 2022), $t_H$ is hour time (0~23), $t_{JD}$ is day of the year (1~365) (Carslaw and Taylor, 2009).

Training datasets of the RF model was conducted on 80% of the original datasets, and the remaining 20% was selected as testing datasets. In the meteorological normalization process of $O_3$ concentration, meteorological variables such as WS, WD, BLH, and cluster, which signify dispersion and transport, were randomly sampled. In the case of $O_3$ precursors, namely $NO_2$ and NMHCs, resampling was exclusively applied to WS, WD and BLH. $NO_2$ and NMHCs have short atmospheric lifetimes, making them less susceptible to the influence of regional transport over large scales (Wang et al., 2023). The resampled specific meteorological variables, along with other initial variables, were fed into the RF



model to predict air pollutants concentrations. The resampling and prediction process were repeated

1000 times to generate 1000 predicted pollutants concentrations. The average values were taken as the

final meteorological normalized concentrations. The RF model was constructed using R"deweather"

packages developed by Carslaw (https://github.com/davidcarslaw/deweather).

### 2.3 Observation-based model

An observation-based model is used in this study to simulate the formation of $O_3$. The model is based

on Regional Atmospheric Chemical Mechanisms version 2 (RACM2) updated with detailed isoprene

oxidation mechanism (Goliff et al., 2013). As a 0-D model, this model incorporates dilution mixing

within the boundary layer. However, vertical or horizontal transport of the air mass is not considered in

this model. Detail of the observation-based box model can be found in Tan et al. (2017). The photolysis

frequencies (J values) were calculated by using the Tropospheric Ultraviolet and Visible (TUV) model

(Wolfe et al., 2016). Model calculations were constrained to measured trace gases, including inorganic

species ($NO_2$ and $O_3$) and organic species (VOCs). Besides, physical parameters like J values,

temperature, pressure and relative humidity were also constrained to measured values. The empirical

kinetic modeling approach (EKMA) serves as a sensitivity test for the OBM. EKMA curve offers a

means to quantify intricate nonlinear relationships among $O_3$, NOx and VOCs, which can be used as a

theoretical basis for designing $O_3$ pollution reduction strategies (Tan et al., 2018). In this study, a total

of 30 emission scenarios were established for both NOx and anthropogenic VOCs. Subsequently, $O_3$

concentrations resulting from changes in these precursor emissions were simulated across 900

scenarios. The EKMA curve was plotted according to the $O_3$ formation rate under different VOCs and

NOx conditions.

### 2.4 Positive matrix factorization


The positive matrix factorization model is based on a large number of data to estimate the compositions

and contributions of emission sources (Paatero and Tapper, 1994). The PMF model is widely used for

VOCs source apportionment (Song et al., 2021; Yuan et al., 2010). In the PMF model, it is assumed

that the pollutants concentrations measured at the receptor point can be represented as a linear sum of

components emitted by different sources. Indeed, the temporal variation of atmospheric pollutants is

influenced not only by emissions but also by dispersion. Direct PMF analysis based on observed data



may lead to the loss of real information regarding emission sources. In this study, the observed and meteorological normalized VOCs concentrations were fed into US EPA PMF v5.0 to identify and quantify major emission sources of VOCs. In contrast to the PMF results based on observation, examining the alterations in contributions of emission sources after meteorological normalization can reveal the impact of dispersion on VOCs sources.

## 3 Results and discussion

### 3.1 Temporal variations of $O_3$ and its precursors

#### 3.1.1 Long-term variations

The RF model demonstrated effective performance in predicting most of the air pollutants. The $R^2$ values of $O_3$, $NO_2$ and NMHCs were 0.88, 0.83 and 0.76, respectively. The $R^2$ values of 81% VOC species were in the range of 0.5 to 0.96, and the $R^2$ values of a few VOC species with low concentrations were lower than 0.4. Fig. 1 displayed the time series of air pollutants concentrations based on observation and meteorological normalization from 2019 to 2022. After meteorological normalization, the concentrations of $O_3$ and its precursors were primarily affected by local factors, including precursors emission and chemical reactions. From a long-term perspective, the trends of air pollutants concentrations after meteorological normalization were consistent with those based on observation. This indicated that the variation in $O_3$ concentration in Hangzhou was mainly driven by precursors emissions and chemical formation in the long term.

From the diurnal trends of $NO_2$ and NMHCs, the observed concentrations were lower during the day and higher at night, which was contrary to the daily trends of WS and BLH (Fig. S1). Stable WS and low BLH at night were not conducive to the diffusion of air pollutants, resulting in the accumulation of pollutants concentrations, while the situation was opposite during the day (Song et al., 2018). After the dispersion effect was removed, the precursors concentrations decreased at night and increased significantly during the day. The diurnal variation of $O_3$ concentration showed a typical single-peak structure before and after meteorological normalization. Different from the change in the concentrations of precursors, the $O_3$ concentration increased at night and decreased during the day after meteorological normalization. At night, the titration reaction of NOx and the horizontal transport reduced the $O_3$ concentration (Li et al., 2022). The NOx concentration decreased after meteorological



normalization, and the weakening of titration resulted in the increase of O₃ concentration at night. In
       addition, the decrease in horizontal transport at night also resulted in the increase of O₃ concentration
       after normalization. During the day, the destruction of the stable boundary layer strengthened the
       vertical mixing effect of the atmosphere, so that the O₃ in the upper atmosphere mixed with the O₃
       generated near the surface, increasing the O₃ concentration (Lei et al., 2023). When the effect of
transport was removed, the daytime O₃ concentration decreased. It can be seen from the diurnal
       variations that meteorological factors directly affected the concentrations of precursors through
       dispersion. And meteorological factors not only directly affected the O₃ concentration through
       horizontal and vertical transport, but also indirectly change O₃ concentration by influencing precursor
       concentration and titration reaction.


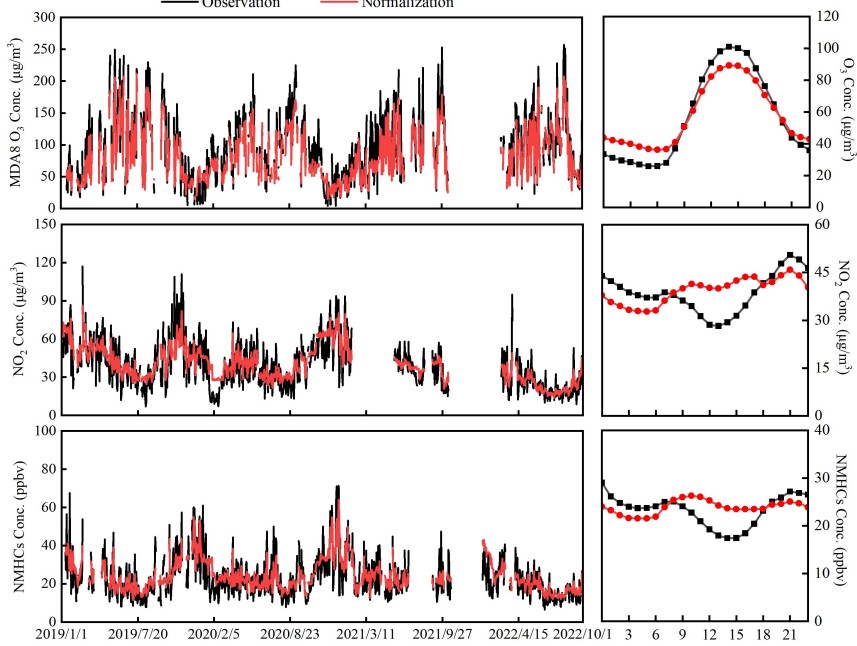

**Figure 1: Long-term trends of daily average concentrations of air pollutants (left) and mean diurnal
variations of air pollutants concentrations (right) based on observation and meteorological normalization
from 2019 to 2022.**


       Fig. 2 showed the importance of the different features in the RF model. The time variables can
       represent anthropogenic emissions to some extent. The chemical reaction of O₃ formation was affected



by meteorological factors such as UVB, T and RH. Local dispersion of $O_3$ and its precursors was mainly affected by WS, WD and BLH, and long-distance transport of $O_3$ was characterized by cluster.

The importance of local chemical reactions to $O_3$ was 83.9%. UVB, influencing photochemical reactions, emerged as the most crucial factor for $O_3$ concentration, with an importance of 25.9%. This is consistent with the findings by WENG et al. (2022) in the same region. Additionally, the importance of RH and T to $O_3$ was also evident, with the importance of 18.2% and 11.3% respectively. Relative humidity was related to cloud cover, exerting an indirect influence on aerosol radiation (Gao et al.,

2021; Ma et al., 2021). Further considering the complex HOx chemical reactions, humidity and $O_3$ concentration were usually negatively correlated (Han et al., 2020). High humidity can enhance the reaction of $O(^1D)$ produced by $O_3$ photolysis and $H_2O$: $O(^1D) + H_2O \rightarrow 2OH$ (Wang et al., 2013). The influence of temperature on $O_3$ formation stemmed from the fact that the chemical kinetic rate increased with rising temperature (Li et al., 2020). Besides, elevated temperature enhanced the

emission of biogenic VOCs (Lu et al., 2019). Hence, some $O_3$ pollution events were associated with high temperature (Dang et al., 2021). Ding et al. (2023) found that temperature was the dominant factor affecting $O_3$ concentration in Tianjin. Wind and BLH also played significant roles in $O_3$ concentration (16.1%), mainly through vertical diffusion, vertical convection and horizontal convection (Li et al., 2012).

Different from $O_3$, BLH exerted a most significant impact on $NO_2$ and NMHCs variation, with the importance value of 26.1% and 20%, respectively. Turbulent mixing in the active boundary layer facilitated the dispersion of air pollutants, whereas the stable boundary layer attenuated vertical diffusion, thereby intensifying the accumulation of air pollutants near the ground. (Huang et al., 2020). The importance of dispersion to $NO_2$ and NMHCs was 34.2% and 30.7, respectively. Consequently,

unfavorable meteorological dispersion conditions can result in the accumulation of precursors, causing $O_3$ pollution even in scenarios with low emissions. Temporal variables representing emissions, such as month and day of year, also occupied important positions. The importance of month to $NO_2$ and NMHCs exceeded 18%, which represented the significant influences of seasonal anthropogenic emissions on the concentrations of precursors. The importance of local emission, production and

consumption to $NO_2$ and NMHCs were 65.8% and 69.3%, respective (Fig. 2).





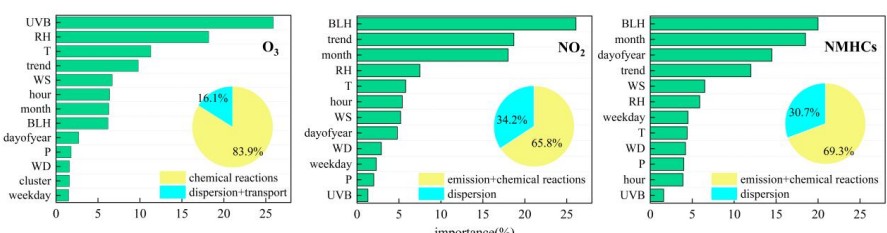

**Figure 2: The importance of each feature to O$_3$, NO$_2$ and NMHCs in the RF model.**

### 3.1.2 Comparison between pollution and non-pollution periods

O$_3$ pollution occur frequently between May and September each year. In order to evaluate the influences of meteorological conditions on the concentrations of O$_3$ and its precursors, the relative change of air pollutants concentrations caused by meteorological factors during O$_3$ pollution and non-pollution periods in warm season from 2019 to 2022 was analyzed. In the non-pollution periods, the negative effect of dispersion on the concentrations of NO$_2$ and NMHCs was apparent, with average relative changes ranging from -9.3% to -27.98% for NO$_2$ and -10.5% to -22.8% for NMHCs. Dispersion and transport have less influences on the MDA8 O$_3$ concentrations, with average relative change ranging from -0.1% to 8.1%. During the pollution periods, the positive effects of dispersion and transport on O$_3$ became evident (from 12.9% to 24.0%). Simultaneously, the negative effect of dispersion on the concentrations of precursors decreased and even transformed into positive effect. Especially in 2021, dispersion had a significant positive effect on NO$_2$ and NMHCs, with an average relative change of 7.8% and 11.8%, respectively. O$_3$ concentration was affected by the long-distance transport as well as the deterioration of diffusion conditions in the pollution periods. Therefore, the influences of meteorological factors on O$_3$ was more obvious than that of its precursors during pollution periods in the warm season.

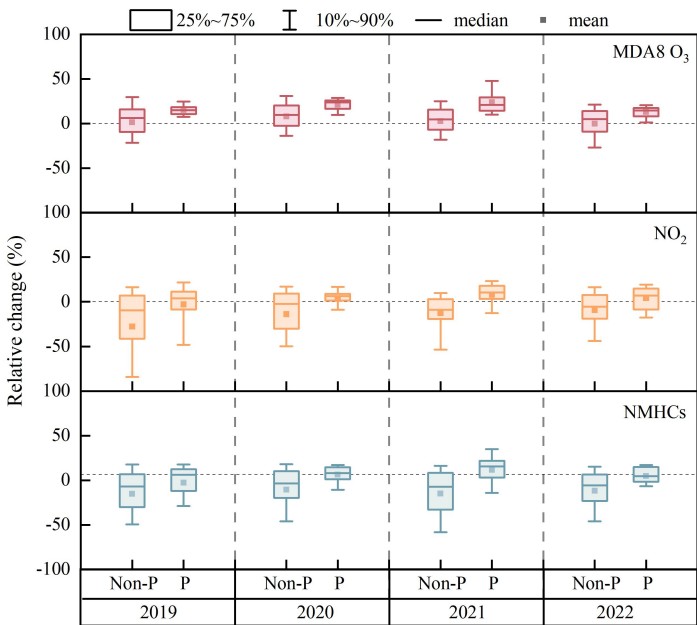

**Figure 3: Relative change caused by meteorological factors during O₃ pollution (P) and non-pollution (Non-P) periods in the warm season from 2019 to 2022, relative change = the observed concentrations - the meteorological normalized concentrations/the observed concentrations.**

### 3.1.3 Variations during short-term pollution events

In order to explore the effects of meteorological dispersion and transport on O₃ concentration in the short term, we selected two typical pollution periods from 2019 to 2022. During the Period 1 (August 31 to September 13 in 2020), the average MDA8 O₃ in Hangzhou was 193 µg/m³ in the pollution, exceeding the national air quality standard (> 160 µg/m³, GB 3095-2012). At the same time, other cities in the YRD regions such as Shanghai, Nanjing, Wuxi, Changzhou, Suzhou and Jiaxing also experienced O₃ pollution (Fig. S2). The Period 1 represented a large-scale regional pollution event. During the pre-pollution (August 31 to September 2 in 2020), dispersion and transport had negative effects on MDA8 O₃. In the pollution periods (September 3 to September 10 in 2020), the concentration of locally generated O₃ (depicted by the red line) remained below the limit, with an average concentration of 157 µg/m³, with only slight exceedances recorded on September 6 and September 9. However, the actual observed O₃ concentration was much higher than the standard, and the O₃ concentration was about 200 µg/m³ from September 6 to September 10. The positive



contributions of dispersion and transport was significant (depicted by the red area) in the pollution

periods, resulting in an 18.7% increase in the MDA8 $O_3$ concentration. During the post-pollution

period, contributions of dispersion and transport decreased significantly.

In the Period 2 (August 10 to August 22 in 2022), the average MDA8 $O_3$ concentration in Hangzhou

was as high as 211 µg/m$^3$ during the pollution, while the concentration of MDA8 $O_3$ in most

surrounding cities was less than 160 µg/m$^3$. Thus the $O_3$ pollution in the Period 2 was influenced by

both local formation and transport. During the pollution periods (August 13 to August 19 in 2022),

locally generated $O_3$ basically exceeded the standard, and the MDA8 $O_3$ concentration was greater than

180 µg/m$^3$ on most days, with an average concentration of 185 µg/m$^3$. On August 16, the

meteorological negative contribution (-14.4%) appeared, exerting dilution effects on the $O_3$

concentration, but the MDA8 $O_3$ on that day still exceeded 160 µg/m$^3$, indicating intense local $O_3$

production. The positive contributions of dispersion and transport to $O_3$ were significant during the

pollution periods, the contributions ranged from 8.5% to 20.4%. For precursors, the concentration of

NMHCs increased between 17 and 19 August (Fig. S3). The positive contribution of dispersion to $NO_2$

and NMHCs ranged from 4.4% to 13.7% and from 0.6% to 8.5% in pollution. During the post-

pollution (August 20 to August 22 in 2022), the contributions of dispersion and transport turned

negative, indicating that meteorological diffusion conditions were in favor to the elimination of $O_3$

pollution.



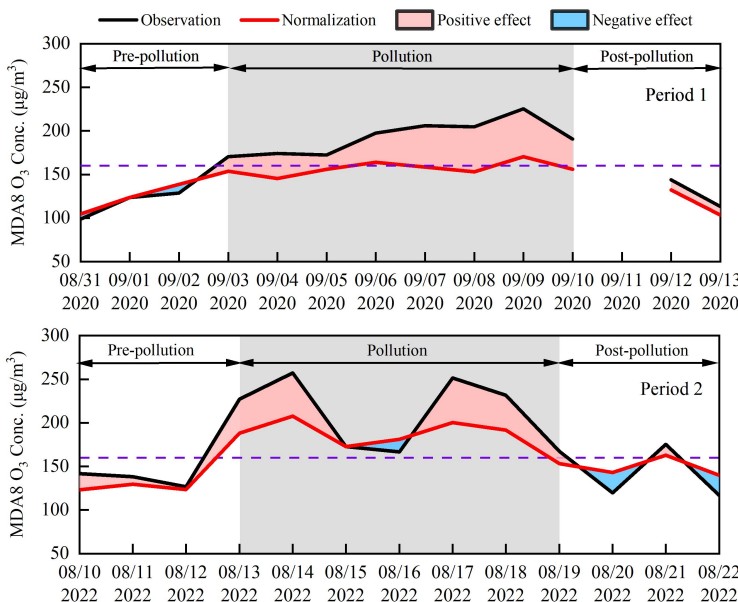

Figure 4: The MDA8 $O_3$ concentration based on observation and meteorological normalization, and the contributions of dispersion and transport to the MDA8 $O_3$ during pre-pollution, pollution and post-pollution in the Period 1 and Period 2 (red: positive contribution, blue: negative contribution).

### 3.2 VOC-NOx-$O_3$ sensitivity

Unfavorable meteorological conditions can cause the accumulation of $O_3$, making it essential to have a clear understanding of local $O_3$ formation pathways for effective control of $O_3$ pollution. The relationship between $O_3$ and $NO_2$ under long-term trends was analyzed based on the observed and meteorological normalized data (Fig. 5). The red dotted line showed the turning point of the relationship between $O_3$ and $NO_2$ concentrations. On the left side of the red dotted line, $O_3$ concentration elevated with the increase of $NO_2$ concentration. At this point, controlling the emission of $NO_2$ was conducive to limiting the formation of $O_3$, suggesting that the sensitivity of $O_3$ formation was limited by NOx. On the right side of the red dotted line, $O_3$ concentration decreased with the increase of $NO_2$ concentration. At this point, the inhibition effect of NOx emission reduction on $O_3$ formation was not significant, and it is necessary to control the emission of VOCs, indicating that the sensitivity of $O_3$ formation was limited by VOCs (Kong et al., 2024). After meteorological normalization, the $NO_2$ concentration in the turning point increased from 9 μg/m³ to 19 μg/m³, suggesting when $NO_2$





concentration was at a higher level, $O_3$ concentration decreased with the increase of $NO_2$ concentration. In other words, the actual $O_3$ production enter the VOC-limited regime more slowly. In addition, based on average results in warm season each year, the sensitivity of $O_3$ formation before and after

325 meteorological normalization was also shown in Fig. 5. Whether based on observed or meteorological normalized data, the $O_3$ formation from 2019 to 2021 was located in the VOC-limited regime, while $O_3$ production enter the transition regime between VOC- and NOx-limited regimes. in 2022.

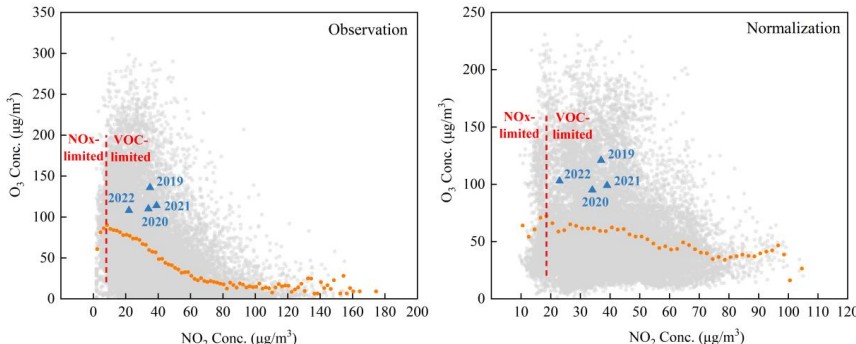

**Figure 5: The changes of $O_3$ concentration on $NO_2$ concentration from 2019 to 2022. The light gray circles represented the hourly $O_3$ concentration. The orange circles represented the average value of $O_3$ concentration in each interval (2 μg/m$^3$ ) of $NO_2$. The blue triangle represented the average value of the MDA8 $O_3$ during the warm season each year.**

The OBM was used to analyze the sensitivity of $O_3$ formation. The OBM is zero-dimensional, meaning it excludes the processes of atmospheric transport and dispersion. Therefore, it is reasonable to remove the influences of transport and dispersion when using the OBM. The VOC-NOx-$O_3$ sensitivity and the net ozone production rate ($P(O_3)$) exhibited significant differences before and after meteorological normalization in the short-term pollution events (Fig. 6). The $O_3$ concentration in the Period 2 was

affected by both transport and local formation. The concentration of local precursors increased after removing the effect of dispersion, resulting in the change of the sensitivity of $O_3$ formation. Based on the observation results, the $O_3$ formation in pollution was located in the NOx-limited regime. After meteorological normalization, $O_3$ formation enter the transition regime between VOC- and NOx-limited regimes. Besides, the meteorological normalized $P(O_3)$ was improved after removing the effect

of transport on $O_3$ concentration. Therefore, when OBM was used to analyze the VOC-NOx-$O_3$



sensitivity, removing the influences of dispersion and transport was beneficial to accurately identify the limited regime of $O_3$ formation.

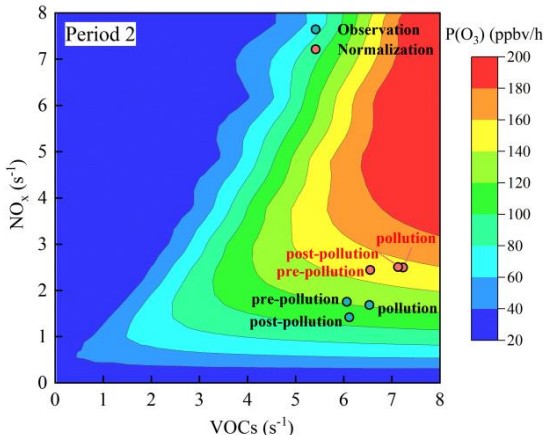

**Figure 6: The $O_3$ isopleth diagram versus NOx and anthropogenic VOCs by using EKMA. The circles represented the average concentrations of NOx and VOC during pre-pollution, pollution and post-pollution in the Period 2.**

### 3.3 VOCs source apportionment

The PMF method was further used for VOCs source analysis. The optimal solution was selected by examining the interpretability of factors and the distribution of scale residuals. Based on observed and meteorological normalized concentrations, seven possible emission sources of VOC from May to September in 2022 were extracted by using the US EPA PMF v5.0. The possible emission sources of VOC included combustion, industrial source, vehicle exhaust, fuel evaporation, secondary and aging source, biogenic source and solvent use. The differences in the source profiles resolved from the observed and normalized concentrations were illustrated in Fig. S4.

Combustion source was characterized by high concentrations of ethane, propane, and acetylene. Low carbon alkanes and alkenes were likely to be the products of incomplete combustion (Wang et al., 2015). Acetylene was a typical tracer of combustion. Toluene and some halohydrocarbons, such as chloromethane, were also released from combustion (Liu et al., 2008). Additionally, the proportion of acetonitrile was also high, which was an important product of biomass combustion (De Gouw et al., 2003). Biomass combustion emission was relatively intense in the YRD. Industrial source was



characterized by halohydrocarbons (Sun et al., 2016), and 1,2-dichloroethane accounted for nearly 80% of this factor in both PMF results. Vehicle exhaust was featured by high concentrations of ethane,

propane, isobutane, n-butane, isopentane, ethylene and toluene(Cai et al., 2010; Liu et al., 2008). Fuel evaporation was characterized by the high concentration and proportion of isopentane, isobutane, n-butane and n-pentane. While the concentration of acetylene was minimal in this factor. Secondary and aging source was characterized by halohydrocarbons and oxygenated VOCs (OVOCs). Methacrolein (MACR) and methyl vinyl ketone (MVK) were products of the oxidation of isoprene (Mo et al., 2018).

OVOC and halohydrocarbons have long lifetimes in the atmosphere and can serve as important tracers for aging sources (Yang et al., 2021). Biogenic source was featured by highest concentration of isoprene, primarily emitted by plants (Gong et al., 2018). Additionally, the oxidation products of isoprene (MACR and MVK) also contributed to this factor. Solvent source was characterized by high concentrations of aromatics. Toluene, ethylbenzene, m-xylene and o-xylene, which were commonly

used as the materials in solvents (Song et al., 2021).

After normalizing the effect of dispersion, the absolute contribution of emission sources to VOCs changed. The mean absolute contribution of vehicle exhaust to VOCs increased most significantly, from 3.97 ppbv to 6.72 ppbv during the non-pollution periods, and from 6.84 ppbv to 9.76 ppbv during the pollution periods. The mean absolute contribution of combustion decreased by 1.55 ppbv and 2.09

ppbv to 2.86 ppbv and 5.85 ppbv during the non-pollution and pollution, respectively. Dispersion caused overestimation of the contribution of combustion to VOCs, which indicated the reduction in VOCs concentration by abatement measures can be offset by the effect of dispersion. Therefore, the impact of dispersion should be taken into account when evaluating the effectiveness of emission reduction measures on VOCs emission sources. The normalized contributions of solvent use and

industrial source in the pollution were comparable, with an average absolute contribution of 2.78 ppbv and 2.57 ppbv. In comparison to the result based on observation, the absolute contribution of fuel evaporation decreased from 1.94 ppbv to 1.33 ppbv after meteorological normalization during the pollution periods. After meteorological normalization, the contributions of biogenic source and secondary and aging source to VOCs during the pollution period were relatively low, with absolute

contributions of 0.54 ppbv.



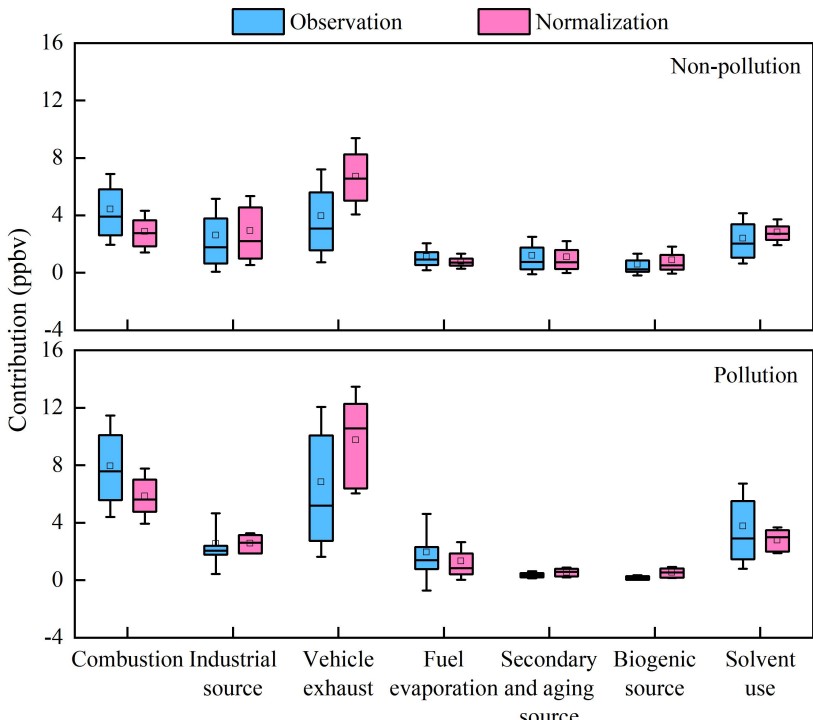

**Figure 7: The absolute contributions of emission sources to VOCs based on observation and meteorological normalization during the non-pollution periods and pollution periods in the warm season in 2022.**


Fig. 8 showed the proportion of VOCs sources before and after meteorological normalization during the non-pollution periods and pollution periods. According to the result of observation, combustion and vehicle exhaust were the largest contributors to VOCs, accounting for 27.1% and 24.3% in the non-pollution periods. And the proportion of combustion and vehicle exhaust increased to 33.7% and 29%

in the pollution periods. During the pollution periods, the proportion of solvent use and fuel evaporation also increased, accounting for 15.9% and 8.2%, respectively. After the normalization of dispersion, vehicle exhaust became the predominant emission source of VOCs (37% in the non-pollution periods and 41.8% in the pollution periods), much higher than the proportion of other emission sources. According to the motor vehicle data released by the Zhejiang Public Security

Department in 2022, the number of motor vehicles reached 23.29 million. During the non-pollution periods, the contributions of solvent use, industrial source and combustion were comparable, accounting for the proportions ranging of 15.6% to 16.2%. However, the influence of combustion on





VOCs increased (25.1%), while the proportion of industrial source and solvent use decreased during the pollution periods (11% and 11.9%). Straw burning occurred frequently in Zhejiang Province.

According to the remote sensing monitoring of straw burning announced by the Ecological Environment Monitoring Center of Zhejiang Province, a total of 135 straw burning points in the province were monitored by satellite remote sensing from January to October 2022. The proportion of industrial emission and solvent use decreased during the pollution periods, indicating that the implementation of shutdown or off-peak production measures at the time of pollution warning were

effective.

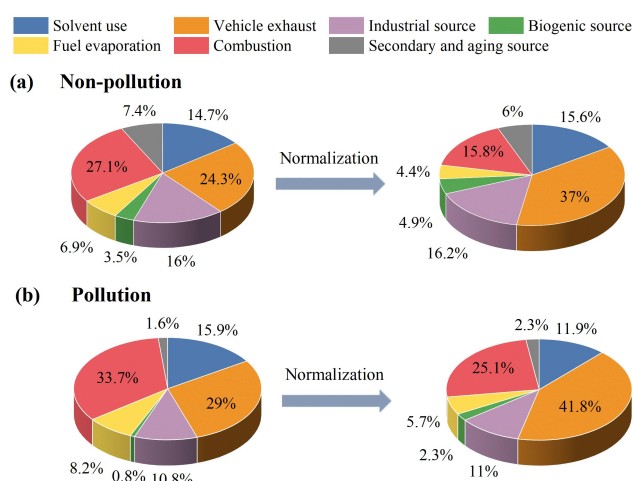

**Figure 8: Comparison of VOCs sources proportion before and after meteorological normalization during the non-pollution periods and pollution periods in the warm season in 2022.**


The O$_3$ formation potential (OFP) is used to assess VOC photochemical activity (Carter, 2010), and it can be calculated by using Eq. (2):

$$OFP_i = MIR_i \times [VOC_i] \tag{2}$$

Where MIR$_i$ represents the maximum incremental reactivity for VOC species i. [VOC]$_i$ represents the

concentration of VOC species i (μg/m$^3$). MIR value for each VOC species were taken from the updated Carter research results (http://www.engr.ucr.edu/~carter/reactdat.htm, last access: 24 February 2021). The contributions of emission sources to OFP was further analyzed and shown in Fig. 9. Based on the result of the observation, the emission sources that contribute the most to OFP were solvent use (67.79





µg/m$^3$), vehicle exhaust (33.16 µg/m$^3$) and combustion (29.16 µg/m$^3$) during the pollution periods in

the warm season in 2022. After removing the effect of dispersion, the contribution of vehicle exhaust to

OPF increased to 47.25 µg/m$^3$, while the contribution of solvent use and combustion to OFP decreased

to 54.77 µg/m$^3$ and 22.58 µg/m$^3$, respectively. The actual contributions of combustion and solvent use

to O$_3$ formation were larger under dispersion effect. Thus, it was necessary to consider the cumulative

effect of dispersion and enhance emission reduction measures for specific emission sources. For the

Period 2 mentioned in section 3.1.3, we also found that the contributions of VOCs emission sources

changed after meteorological normalization (Fig. S5 and Fig .S6). After removing the dispersion effect,

the contributions of solvent use and vehicle exhaust to OFP increased during the pollution, while the

contribution of combustion and secondary and aging source decreased. From August 17 to August 19,

the normalized contribution of solvent source to OFP was significant, with an average OFP of 105.81

µg/m$^3$, indicating that the emission of solvent source was enhanced in these days. The dispersion effect

of meteorological conditions on precursors can conceal the real information of emission sources and

misjudge the formation process of O$_3$.

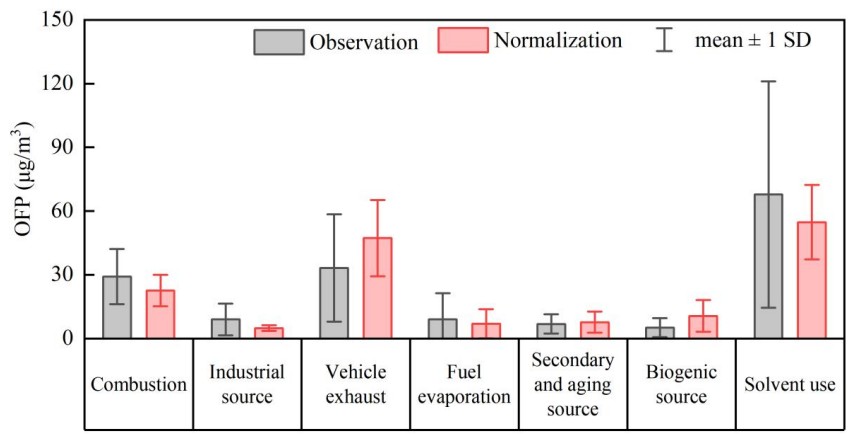

**Figure 9: The contributions of emission sources to OFP based on observation and meteorological normalization during the pollution periods in the warm season in 2022.**



## 4 Conclusion

In this paper, a RF model was established based on the hourly data of four years of continuous

observation, and some meteorological effects on the concentration time series of air pollutants were

removed. Transport and dispersion effects were removed for $O_3$ and dispersion effect was removed for

its precursors. In the process of building the RF model, UVB, RH and T were found to be the most

important factors affecting $O_3$ concentration, with the importance of 25.9%, 18.2% and 11.3%,

respectively. Local influences, including precursor emissions and secondary photochemical reactions,

occupied 83.9% of the importance to $O_3$ concentration. To understand the mechanisms of local $O_3$

formation, the meteorological effects were analyzed in long-term trends, pollution and non-pollution

periods in the warm season, as well as short-term pollution events. After decoupling meteorological

effects, the concentration trends of $O_3$ was consistent with those observed in the long term, indicating

that $O_3$ concentration was mainly driven by precursor emissions and local chemical reactions. During

the pollution periods in the warm season from 2019 to 2022, the positive contributions of dispersion

and transport to the MDA8 $O_3$ ranged from 12.9% to 24.0%. The effects of dispersion and transport

were further analyzed for different types of $O_3$ pollution events. For transmission-type $O_3$ pollution

(Period 1), dispersion and transport contributed 18.7% to the MDA8 $O_3$ concentration, increasing the

mean MDA8 $O_3$ concentration from 157 μg/m$^3$ to 193 μg/m$^3$. For local and transmission-type $O_3$

pollution (Period 2), the average locally generated MDA8 $O_3$ concentration was 185 μg/m$^3$. Under the

influences of dispersion and transport, the average MDA8 $O_3$ concentration increased to 211 μg/m$^3$,

and the positive contributions of dispersion and transport ranged from 8.5% to 20.4%. BLH, as a

parameter of dispersion, was of the highest importance for $NO_2$ and NMHCs, accounting for 34.2% to

$NO_2$ and 30.7% to NMHCs. Therefore, precursor concentrations were accumulated even in the case of

low emissions when the dispersion condition was poor, promoting the photochemical production of $O_3$.

This also corresponds to the fact that even with the implementation of precursor emission reduction

policies, $O_3$ concentrations in urban areas remain persistently high.

By decoupling the influences of meteorological conditions, it was observed that the sensitivity of local

$O_3$ formation and the apportionment of VOCs emission sources have changed. From the EKMA of

short-term pollution event, the sensitivity of $O_3$ formation in Period 2 changed from the NOx-limited

regime to the transition regime between VOC- and NOx-limited regimes after meteorological



normalization. Based on PMF model, the changes of VOCs emission sources after the removal of dispersion effect during the warm season in 2022 were further analyzed. After removing the effect of dispersion, the absolute contribution of vehicle exhaust to VOCs during the pollution was 9.76 ppbv,

accounting for 41.8%, and the contribution of vehicle exhaust to OFP was 47.25 μg/m³. The contribution of vehicle exhaust to VOCs was underestimated due the dispersion effect. After meteorological normalization, combustion remained an important source of VOCs, contributing 25.1% during the pollution period, which was overestimated by 8.6%. The normalized contribution of solvent use to VOCs decreased to 11.9%, but it is undeniable that solvent use was still a crucial contributor to

OFP, contributing 54.77 μg/m³. Neglecting the influences of meteorology can lead to a deviation in emission reduction priorities, and the effectiveness of emission reduction may be masked by unfavorable meteorological conditions. The conclusion of this research suggested that meteorological fluctuations can interfere with the results of OBM and PMF. Decoupling meteorological effects before using traditional models was beneficial for deepening the understanding of local $O_3$ formation and

improving the rationality of precursor emission reduction measures.



**Data availability.** The data used in this study are available upon request from Yuqing Qiu (yuqing.qiu@stu.pku.edu.cn) and Xin Li (li_xin@pku.edu.cn).

**Author contributions.** XL, WC, and YZ conceived and designed this study, and revised the Article critically. YQ and XL analysed and interpreted data, drafted the Article, and revised it critically. YL 500 and MS contributed to the modeling of the data. XT, QZ, WL, WZ, and JL acquired the field observation data.

**Competing interests.** The authors declare that they have no conflict of interest.

**Acknowledgements.** The authors are grateful to the Zhejiang Ecological and Environmental Monitoring Center and Jinhua Ecological and Environmental Monitoring Center for observation in this 505 study. This work was supported by the Beijing Municipal Natural Science Fund (JQ21030) and by the National Key R&D Program of China (2022YFC3700302).

**Financial support.** This research has been supported by the Beijing Municipal Natural Science Fund (JQ21030) and the National Key R&D Program of China (2022YFC3700302).



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
