# Peer review of "Insights on ozone pollution control in urban areas by decoupling meteorological factors based on machine learning"

_EGUsphere, 2024_

## Referee Comment (RC1)

The manuscript titled "**Insights on ozone pollution control in urban areas by decoupling meteorological factors based on machine learning**" uses a machine learning method to decouple the meteorological effects on concentrations of $O_3$ and its precursors. This method provides better understanding of $O_3$ precursor sensitivity and sources of VOCs. The article is well organized. It can be accepted after considering the following suggestions.

Line 15: The term "atmospheric environment capacity" sounds weird.

Line 81-83: This statement is not justified. The emission reduction policies do not necessarily rely on the two methods, and other methods such as air quality model can also provide basis.

Lines 107-109: How to measure these meteorological factors should be given.

Line 130: the "cluster" should be explained here.

Lines 131-132: From my understanding, the variable "trend" just characterize the date and hour. Why does it relate to the implementation of policy measures?

Line 137-138: It is better to randomly split the data into ten subsets, and randomly use nine of them for training and the rest one for testing.

Line 125: For the performance of the random forest model, which variables are response variable and which are predictors should be clarified.

Line 141: Different VOCs species has different lifetime. Some VOCs with low reactivity have longer lifetime, which can go through regional transport. The difference among different VOCs species should be considered.

Lines 144-145: Which time periods are selected for the resampling? The whole four years or the month to which the investigated day belongs to? This should be clarified.

Line 187: I suggest to give some quantitative description of the consistency.

Line 190: "From the diurnal trends of NO2 and NMHCs," sounds weird.

Line 207: "And" is redundant.

Lines 205-210: $O_3$ concentrations can affect the nighttime NO2 and VOCs by titration and ozonolysis reactions of alkenes. How do you evaluate it?

Fig. 2. How to evaluate the importance of different features should be depicted in the Method.

Line 215: the reason why the time variables can represent anthropogenic emissions should be clarified.

Line 226-227: This reaction will cause more production of OH, which will increase O3 production. So this probably cannot explain the negative correlation between RH and O3. Higher RH generally corresponds to more cloud and precipitation, causing lower O3 concentrations. The reason for the negative correlation should be double checked.

Line 227-229: In fact, reaction rates does not necessarily increased with temperature increasing. In fact, many important reactions such as NO2+OH and some VOCs+OH will get slower with higher temperature.

Temperature not only affects chemical reactions and precursors emissions, but also affects physical processes. How do you isolate the physical effects?

Line 281: The term "locally generated O3" should be defined or explained here.

Line 323: "…more slowly" this description is not clear. Higher value of the turning point indicates the real NOx concentrations is more likely lower than this value, suggesting a higher possibility to be in the NOx-limited regime.

Line 327: The transitional regime is not defined here. Do you mean the turning points is transitional regime?

Figure 5. The relationship between O3 and NO2 and the turning point are acquired from the normalized O3 and NO2. However, it seems that the average values of NO2 for each year are acquired from the observed values, rather than the normalized values. The reason for the inconsistency should be clarified.

Figure 5 and Figure 6. In Figure 5 O3 sensitivity shifts from a VOC-limited regime to a NOx-limited regime, while in Figure 6, this shift is toward inverse direction. The contradiction should be explained.

Figure 6. How do you judge that O3 sensitivity shifts from NOX-limited regime to transition regime? It seems that it is in a NOx-limited regime for both cases.

Line 344: "besides, …." This sentence is unclear to me.

Lines 412-414: It is unclear what the decrease or increase of VOCs is relative to. Is it relative to non-pollution period, or observed concentrations?

417-420: Here, you state that the proportion of industrial emission and solvent use decreased. This does not mean the concentrations of VOCs decrease. So this cannot demonstrate the shutdown measures are effective. I suggest to additionally show the changes of VOCs concentrations from different sources in this Figure or in supplementary materials.

---

## Referee Comment (RC2)

The manuscript by Qiu et al. investigates the sources and meteorological factors influencing ozone variation over four years in Hangzhou China, using observation-based approaches including machine learning (ML) -based meteorological normalization, PMF, and a Box Model for ozone simulation. Overall, the manuscript is well-organized, clearly written, and presents the results effectively. The application of ML in this study provides a strong example of its potential to enhance our understanding of ozone formation. My comments below are primarily focused on the methodology regarding source apportionment and the ML aspects, which the authors identify as novel points of this work.

**General comments**

Ozone concentrations are determined by various drivers (e.g., precursor emissions, dilution, transport, deposition, and chemistry). It is easy to relate ambient ozone to its drivers using ML algorithms, while it is important to emphasize the physical interpretation, not just the mathematical relationships, in data-driven approaches. This is why knowledge-guided ML is now highly recommended. Specifically, in the application of ML for explaining ozone formation, emphasis should be placed on feature selection (i.e., variables representing potential drivers) and the interpretation of results.

1. In the Methods section, the meteorological normalization method is applied to decouple the impact of meteorology from emission-driven changes in ozone and source-specific VOCs. ML-based meteorological normalization is essentially an adjustment method that aims to correct meteorologically induced variations in air quality time series. Similar statistical approaches have been used since the 1980s in the USA to estimate emission-driven trends of ozone. It is important to clarify that this technique does not "remove" meteorology from observational data but rather reduces its impact through specific techniques. We cannot have air pollution without meteorology. The term "REMOVE" is used throughout the text, it would be prudent to use quotation marks around "REMOVE" to avoid misunderstanding.

2. A key question here is the physical meaning of meteorologically normalized ozone. The level of normalized air pollutants depends on how normalization is applied according to the research purpose. Section 2.2 focuses heavily on random forest modeling but lacks sufficient detail about the rationale for feature selection and the meteorological normalization processes, making it difficult for readers to fully understand the implications of the results. The authors discuss the relative importance of dispersion +/ transport and chemistry in driving air pollutants, assuming these atmospheric processes are well represented by variables like wind and air mass clusters. This assumption needs clarification to build confidence in the model results—specifically, what features are proxies for specific atmospheric processes?

3. In Section 2.4, the authors state "*In this study, the observed and meteorological normalized VOCs concentrations were fed into US EPA PMF v5.0 to identify and quantify major emission sources of VOCs.*" This approach is interesting for PMF modeling, particularly in examining changes in source contributions after meteorological normalization to understand the impact of dispersion (should be the overall impact of meteorology) on VOC sources (a good point to address). However, since PMF is a bilinear model requiring additive input variables, questions arise: are these normalized VOCs still additive? How is the total VOC for normalized

concentrations calculated? Is the normalized VOC comparable to the observed VOC? An alternative approach to achieve the same goal might be to meteorologically normalize the PMF-resolved source-specific VOCs (i.e., run PMF with observed VOCs first, then normalize each source-specific VOC). This work may be of the authors interest: https://doi.org/10.1029/2023JD038696.

4. In Figure 2, all features are ranked with positive values, which describe the magnitude of their impacts without considering the sign of those impacts. However, dispersion can have both positive (enhancing concentration during poor dilution) and negative (reducing pollutant levels) effects. Additionally, using pie charts to illustrate the roles of dispersion and chemistry is problematic because chemistry is not independent of dispersion and transport. Can the authors elaborate more about this?

5. In the Results & Discussion section, the authors demonstrate model performance using only the squared correlation coefficients. It is recommended to also include root mean squared errors, as this is an important metric for describing the accuracy of model predictions.

In summary, I strongly recommend that the authors add more details about feature selection, the adopted meteorological normalization process, and the physical meaning of the normalized air pollutants. One of the existing literature has discussed and reviewed various meteorological normalization strategies based on ML modeling, which may be helpful for this work (https://doi.org/10.1007/s11430-022-1128-1).

**Minor Comments**

1. Line 381: Clarify what is meant by "After normalizing the effect of dispersion." Meteorological normalization is not limited to normalizing the effect of dispersion.

2. Figure 8: Regarding the pies for normalized source contributions, are these contributions additive? What is the physical meaning of the sum of normalized source contributions?

---

## Author Comment (AC1)

**Response to Reviewer #1**

***General comments***

*The manuscript titled "Insights on ozone pollution control in urban areas by decoupling meteorological factors based on machine learning" uses a machine learning method to decouple the meteorological effects on concentrations of $O_3$ and its precursors. This method provides better understanding of $O_3$ precursor sensitivity and sources of VOCs. The article is well organized. It can be accepted after considering the following suggestions.*

**Response:**

We would like to express our gratitude to Reviewer #1 for their thorough review of our manuscript and for their valuable and constructive comments. We have carefully revised the manuscript in accordance with the reviewer's suggestions. Below, we provide point-by-point responses to the reviewer's comments. The reviewer's questions are presented in italics, while our responses are in standard font. The corresponding revisions to the manuscript are highlighted in blue. All changes to the original submission are tracked in the revised manuscript. Finally, we would like to once again thank the reviewer for their positive remarks.

***Comments***

*1. Line 15: The term "atmospheric environment capacity" sounds weird.*

**Response:**

We apologize for the unclear wording in our previous statement. What we intended to express was *"atmospheric capacity"* or *"air quality capacity"*, referring to the maximum amount of pollutants that the atmosphere in a specific area can purify. We've revised the following sentence for a more natural and widely accepted phrasing:

When meteorological conditions deteriorate, the atmosphere's capacity to cleanse pollutants decreases, leading to the accumulation of air pollutants.

*2. Line 81-83: This statement is not justified. The emission reduction policies do not necessarily rely on the two methods, and other methods such as air quality model can also provide basis.*

**Response:**

We apologize for the previous overly definitive statement and have revised the paragraph as follows:

The observation-based model (OBM), positive matrix factorization model (PMF), and air quality model are commonly used to analyze the causes of $O_3$ pollution and provide theoretical support for reducing $O_3$ precursors. However, the results of OBM and PMF, which rely on observed data, may be influenced by fluctuations in meteorological conditions, potentially introducing bias.

*3. Lines 107-109: How to measure these meteorological factors should be given.*

**Response:**

Thanks to the reviewer's comments, we have added the measurement information of meteorological factors:

The meteorological parameters measured included temperature (T), relative humidity (RH), atmospheric pressure (P), wind speed (WS), and wind direction (WD), which were measured by the WS500-UMB instrument manufactured by LUFFT Corporation.

*4. Line 130: the "cluster" should be explained here.*

**Response:**

We appreciate the reviewer's comments. We have added the following description when we first mention "cluster" on line 115:

Cluster of backward trajectories were widely employed to represent the main directions of air masses at monitoring sites (Song et al., 2021).

*5. Lines 131-132: From my understanding, the variable "trend" just characterize the date and hour. Why does it relate to the implementation of policy measures?*

**Response:**

We appreciate the reviewer's comments. We used the term "trend" to capture the long-term changes in emission sources, which were closely related to activity levels. Environmental regulations and policies aimed at reducing pollutant emissions were implemented during specific time periods, and the effects of these measures required time to manifest. Therefore, the "trend" not only reflected the changes in emission source intensity but also represented the long-term variation in air pollutants caused by the enforcement of policies and regulations. (Carslaw and Taylor, 2009; Vu et al., 2019). We have added the following description in the manuscript:

The parameter 'trend' can indicate the long-term changes of air pollutants concentrations resulting from the implementation of policy measures (Vu et al., 2019). Environmental regulations and policies aimed at reducing pollutant emissions were implemented during specific periods, and their effects became apparent in the long-term trends. Therefore, the "trend" not only reflected changes in emission sources closely related to activity levels but also represented the long-term variations in air pollutants caused by the enforcement of policies and regulations. The parameter 'trend' was calculated as Eq. (1):

*6. Line 137-138: It is better to randomly split the data into ten subsets, and randomly use nine of them for training and the rest one for testing.*

**Response:**

Thanks to the reviewer's comments. In this paper, we used 80% of the data as the training set and 20% as the testing set. This method has been widely applied in many studies and has proven to be effective (Grange et al., 2018; Liu et al., 2022a). The 10-fold cross-validation you mentioned could further enhance the reliability of model evaluation, and we will consider using this approach for data analysis in future research. Thank you once again for your suggestion.

*7. Line 125: For the performance of the random forest model, which variables are response variable and which are predictors should be clarified.*

**Response:**

We appreciate the reviewer's comments. We have specified which variables are response variables and which are predictor variables:

In the RF model, the air pollutants were the response variables, while the explanatory variables included time variables representing source emissions and meteorological variables representing physical and chemical processes.

*8. Line 141: Different VOCs species has different lifetime. Some VOCs with low reactivity have longer*

*lifetime, which can go through regional transport. The difference among different VOCs species should be considered.*

**Response:**

Thanks to the reviewer's comments The point you raised about the varying lifetimes of different VOCs is indeed important. We included the cluster representing long-range transport as an explanatory variable in the RF model for NMHCs and several VOCs with different lifetimes. The results showed that the feature importance of the cluster was the lowest, ranging from 0.5% to 1.2%. The importance of cluster for acetylene, a long-lived compound, was 1.2%, while for ethylene, a short-lived compound, it was 0.5%. For NMHCs, the cluster importance was 1%. Compared to $O_3$, the impact of the cluster on VOCs was insignificant. The uncertainty of the cluster's impact on VOCs was approximately 1%. Within this margin of error, we approximated that VOCs were primarily influenced by dispersion effects. We have added the following clarification to the manuscript:

To take into consideration that some NMHCs have relatively long lifetimes (such as acetylene), the cluster was incorporated as an explanatory variable in the RF model. For NMHCs with different lifetimes, the feature importance of the cluster was relatively low (around 1%). Therefore, it can be approximated that NMHCs were primarily influenced by dispersion effects within the uncertainty.

*9. Lines 144-145: Which time periods are selected for the resampling? The whole four years or the month to which the investigated day belongs to? This should be clarified.*

**Response:**

We appreciate the reviewer's suggestion. We have clarified the resampling period, and the revised sentence is as follows:

The resampling of meteorological variables was conducted over the two-week period before and after the selected date, with the resampled hours remaining constant. This approach effectively preserved the seasonal and diurnal variations in the response variables (Vu et al., 2019).

*10. Line 187: I suggest to give some quantitative description of the consistency.*

**Response:**

We appreciate the reviewer's suggestion. We have given some quantitative description of the consistency, and the revised sentence is as follows:

After meteorological normalization, MDA8 $O_3$ significantly decreased in 2020, followed by a slight increase in 2021 and 2022. The observed annual variation in MDA8 $O_3$ exhibited a similar trend. The meteorologically normalized annual mean MDA8 $O_3$ in 2020 decreased by 10% compared to 2019, which aligned with the observed change of -8.7%. Based on both meteorologically normalized and observed results, the concentrations of $NO_2$ and NMHCs showed declining trends, with a significant decrease in 2022. Compared to 2019, the meteorologically normalized concentrations of $NO_2$ and NMHCs in 2022 decreased by 46.1% and 24%, respectively, while the observed concentrations of $NO_2$ and NMHCs decreased by 45.7% and 16%, respectively.

*11. Line 190: "From the diurnal trends of $NO_2$ and NMHCs," sounds weird.*

**Response:**

We appreciate the reviewer's comments. We have revised the sentence to the following:

From the diurnal variation of $NO_2$ and NMHCs concentrations.

*12. Line 207: "And" is redundant.*

**Response:**

We appreciate the reviewer's comments. We have deleted it.

*13. Lines 205-210: O₃ concentrations can affect the nighttime NO₂ and VOCs by titration and ozonolysis reactions of alkenes. How do you evaluate it?*

**Response:**

We appreciate the reviewer's comments. $O_3$ can react with NO to produce $NO_2$, which leads to an increase in nighttime $NO_2$ concentrations. From the diurnal variation of observed $O_3$ and $NO_2$, it can be seen that the nighttime $O_3$ concentration decreased, the corresponding $NO_2$ concentration increased. High concentrations of alkenes can produce Criegee intermediates (CIs) through ozonolysis, which can rapidly decompose into a large number of radicals, facilitating the oxidation of VOCs and participating in radical cycling reactions, ultimately promoting $O_3$ formation. However, this reaction is more importance in areas with high alkene emissions, such as petrochemical regions (Yang et al., 2024). In this study, the concentration of alkenes is below 10 ppb, so the impact of this reaction is minimal.

*14. Fig. 2. How to evaluate the importance of different features should be depicted in the Method.*

**Response:**

We appreciate the reviewer's suggestion. We have added the following explanation in the Method:

Feature importance was used to reflect the overall significance of explanatory variables in the RF model. The importance was typically represented as an array, where each value corresponded to the importance score of a specific feature. These scores usually range from 0 to 1. The higher importance score indicated that the feature had a stronger predictive capability for the response variable.

*15. Line 215: the reason why the time variables can represent anthropogenic emissions should be clarified.*

**Response:**

We appreciate the reviewer's comments. We have added the following clarification:

Time variables were closely related to the periodic changes in human activities. For example, weekdays versus weekends and peak versus non-peak hours corresponded to different levels of anthropogenic emissions. Anthropogenic emissions influenced the seasonal variations of atmospheric pollutants, as seen in winter heating effects. Previous studies also used time variables to represent anthropogenic emissions (Dai et al., 2023; Vu et al., 2019).

*16. Line 226-227: This reaction will cause more production of OH, which will increase O₃ production. So this probably cannot explain the negative correlation between RH and O₃. Higher RH generally corresponds to more cloud and precipitation, causing lower O₃ concentrations. The reason for the negative correlation should be double checked.*

**Response:**

We appreciate the reviewer for pointing out this key issue. We have removed the explanation related to "HOx chemical reactions" and revised it to the following statement:

Higher relative humidity was usually associated with a higher cloud cover, and relative humidity was generally negatively correlated with $O_3$ (Liu et al., 2023).

*17. Line 227-229: In fact, reaction rates does not necessarily increased with temperature increasing. In fact, many important reactions such as NO₂+OH and some VOCs+OH will get slower with higher temperature.*

*Temperature not only affects chemical reactions and precursors emissions, but also affects physical processes. How do you isolate the physical effects?*

**Response:**

We apologize for the unclear expression. We have revised it to the following statement:

High temperatures increased the rate of most chemical reactions in the atmosphere, especially photochemical reactions that lead to $O_3$ formation.

Additionally, the reviewer's important point that "Temperature not only affects chemical reactions and precursor emissions, but also affects physical processes" is very meaningful. In this study, temperature was primarily used as an indicator of chemical reactions. In our machine learning approach, we used the parameters that more directly affect physical processes, such as WS, WD, and BLH. Therefore, we mainly considered the influence of temperature on chemical reactions. The reviewer's question is something we should further contemplate. Thank you once again for your comments.

*18. Line 281: The term "locally generated $O_3$" should be defined or explained here.*

**Response:**

We appreciate the reviewer's comments. We have included the definition of locally generated $O_3$.

Locally generated $O_3$ was produced in the atmosphere through photochemical reactions involving VOCs and nitrogen oxides NOx (Song et al., 2021).

*19. Line 323: "...more slowly" this description is not clear. Higher value of the turning point indicates the real NOx concentrations is more likely lower than this value, suggesting a higher possibility to be in the NOx-limited regime.*

**Response:**

We appreciate the reviewer's comments. We have revised the sentence as follows:

In other words, a higher $NO_2$ value at the turning point suggested a greater likelihood that the actual NOx concentration was below that value, indicating a higher probability of being in a NOx-limited regime.

*20. Line 327: The transitional regime is not defined here. Do you mean the turning points is transitional regime?*

**Response:**

We appreciate the reviewer's comments. We have added the definition as follows:

The transition regime referred to the region near the turning point, where $O_3$ formation was sensitive to changes in both VOCs and NOx.

*21. Figure 5. The relationship between $O_3$ and $NO_2$ and the turning point are acquired from the normalized $O_3$ and $NO_2$. However, it seems that the average values of $NO_2$ for each year are acquired from the observed values, rather than the normalized values. The reason for the inconsistency should be clarified.*

**Response:**

We appreciate the reviewer's comments. We apologize for any misunderstanding. In lines 312-314, we

described the data sources for Figure 5. All the data in the left panel of Figure 5 were based on observed data, including the annual average $O_3$ values (blue triangles). While all the data in the right panel were derived from meteorologically normalized data, with the annual average $O_3$ values also calculated from the meteorologically normalized data. The small differences observed after annual averaging may have contributed to the misunderstanding. We have further added the following description:

The relationship between $O_3$ and $NO_2$ under long-term trends was analyzed based on the observed (left) and meteorologically normalized (right) data (Fig. 5). The red dotted line showed the turning point of the relationship between $O_3$ and $NO_2$ concentrations. The blue triangle represented the average value of the MDA8 $O_3$ during the warm season each year.

*22. Figure 5 and Figure 6. In Figure 5 $O_3$ sensitivity shifts from a VOC-limited regime to a NOx-limited regime, while in Figure 6, this shift is toward inverse direction. The contradiction should be explained.*

**Response:**

We appreciate the reviewer's comments. Figure 5 presented a long-term analysis of $O_3$ formation sensitivity, showing that the annual $O_3$ formation sensitivity located in the VOC-limited regime after meteorological normalization. The corresponding $NO_2$ concentration is higher when entering the VOC-limited regime, indicating that under low NOx conditions, it continued to be in the NOx-limited regime. Figure 6 illustrated the $O_3$ sensitivity analysis during a pollution event, revealing that the $O_3$ formation sensitivity during the event shifted towards the transition regime between VOC- and NOx-limited regimes after meteorological normalization. This implied that during $O_3$ pollution event, coordinated control of both VOCs and NOx was necessary. $O_3$ formation sensitivity varied between long-term and short-term pollution events.

*23. Figure 6. How do you judge that $O_3$ sensitivity shifts from NOx-limited regime to transition regime? It seems that it is in a NOx-limited regime for both cases.*

**Response:**

We appreciate the reviewer's comments.We have made the following adjustments:

Based on the observation results, the $O_3$ formation in pollution was located in the strict NOx-limited regime. After meteorological normalization, $O_3$ formation shifted towards the transition regime between VOC- and NOx-limited regimes. The limitation of $O_3$ formation by NOx concentration was weakened.

[Figure]

**Figure 6: The O₃ isopleth diagram versus NOx and anthropogenic VOCs by using EKMA. The circles represented the average concentrations of NOx and VOC during pre-pollution, pollution and post-pollution in the Period 2.**

*24. Line 344: "besides, ...." This sentence is unclear to me.*

**Response:**

We apologize for any lack of clarity in our expression. We intended to convey that after meteorological normalization, the ozone formation rate $P(O_3)$ increased. As shown in Figure 6, the red points are located in a yellow background, corresponding to higher $P(O_3)$ values. We have revised the statement as follows:

After removing the influence of dispersion and transport on $O_3$ concentrations, the value of $P(O_3)$ increased, indicating that the $P(O_3)$ calculated based on observation was likely underestimated.

*25. Lines 412-414: It is unclear what the decrease or increase of VOCs is relative to. Is it relative to non-pollution period, or observed concentrations?*

**Response:**

We apologize for any unclear expression. We have revised the statement as follows:

During the non-pollution periods, the contributions of solvent use, industrial source and combustion were comparable, accounting for the proportions ranging of 15.6% to 16.2%. Compared to the non-pollution periods, the influence of combustion on VOCs increased (25.1%), while the proportion of industrial source and solvent use decreased during the pollution periods (11% and 11.9%).

*26. Lines 417-420: Here, you state that the proportion of industrial emission and solvent use decreased. This does not mean the concentrations of VOCs decrease. So this cannot demonstrate the shutdown measures are effective. I suggest to additionally show the changes of VOCs concentrations from different sources in this Figure or in supplementary materials.*

**Response:**

We appreciate the reviewer's comments. We have shown the variation of VOC concentrations from different sources in Figure 7, where it can be observed that VOCs from industrial emissions and solvent

usage have decreased. We will add the following explanation:

The proportion of industrial emissions and solvent usage decreased during the pollution periods, and the VOC concentrations from these two sources also declined (Fig. 7), indicating that the shutdown or off-peak production measures implemented during pollution warnings were effective in controlling emissions from these sources.

**References**

Carslaw, D. C. and Taylor, P. J.: Analysis of air pollution data at a mixed source location using boosted regression trees, Atmospheric Environment, 43, 3563-3570, 10.1016/j.atmosenv.2009.04.001, 2009.

Dai, Q., Dai, T., Hou, L., Li, L., Bi, X., Zhang, Y., and Feng, Y.: Quantifying the impacts of emissions and meteorology on the interannual variations of air pollutants in major Chinese cities from 2015 to 2021, Science China Earth Sciences, 66, 1725-1737, 10.1007/s11430-022-1128-1, 2023.

Grange, S. K., Carslaw, D. C., Lewis, A. C., Boleti, E., and Hueglin, C.: Random forest meteorological normalisation models for Swiss PM10 trend analysis, Atmospheric Chemistry and Physics, 18, 6223-6239, 10.5194/acp-18-6223-2018, 2018.

Liu, B., Wang, Y., Meng, H., Dai, Q., Diao, L., Wu, J., Shi, L., Wang, J., Zhang, Y., and Feng, Y.: Dramatic changes in atmospheric pollution source contributions for a coastal megacity in northern China from 2011 to 2020, Atmospheric Chemistry and Physics, 22, 8597-8615, 10.5194/acp-22-8597-2022, 2022.

Liu, Y., Geng, G., Cheng, J., Liu, Y., Xiao, Q., Liu, L., Shi, Q., Tong, D., He, K., and Zhang, Q.: Drivers of Increasing Ozone during the Two Phases of Clean Air Actions in China 2013–2020, Environmental Science & Technology, 57, 8954-8964, 10.1021/acs.est.3c00054, 2023.

Song, M., Li, X., Yang, S., Yu, X., Zhou, S., Yang, Y., Chen, S., Dong, H., Liao, K., Chen, Q., Lu, K., Zhang, N., Cao, J., Zeng, L., and Zhang, Y.: Spatiotemporal variation, sources, and secondary transformation potential of volatile organic compounds in Xi'an, China, Atmospheric Chemistry and Physics, 21, 4939-4958, 10.5194/acp-21-4939-2021, 2021.

Vu, T. V., Shi, Z., Cheng, J., Zhang, Q., He, K., Wang, S., and Harrison, R. M.: Assessing the impact of clean air action on air quality trends in Beijing using a machine learning technique, Atmospheric Chemistry and Physics, 19, 11303-11314, 10.5194/acp-19-11303-2019, 2019.

Yang, J., Zeren, Y., Guo, H., Wang, Y., Lyu, X., Zhou, B., Gao, H., Yao, D., Wang, Z., Zhao, S., Li, J., and Zhang, G.: Wintertime ozone surges: The critical role of alkene ozonolysis, Environmental Science and Ecotechnology, 22, 10.1016/j.ese.2024.100477, 2024.

---

## Author Comment (AC2)

**Response to Reviewer #2**

*The manuscript by Qiu et al. investigates the sources and meteorological factors influencing ozone variation over four years in Hangzhou China, using observation-based approaches including machine learning (ML) -based meteorological normalization, PMF, and a Box Model for ozone simulation. Overall, the manuscript is well-organized, clearly written, and presents the results effectively. The application of ML in this study provides a strong example of its potential to enhance our understanding of ozone formation. My comments below are primarily focused on the methodology regarding source apportionment and the ML aspects, which the authors identify as novel points of this work.*

*General comments*

*Ozone concentrations are determined by various drivers (e.g., precursor emissions, dilution, transport, deposition, and chemistry). It is easy to relate ambient ozone to its drivers using ML algorithms, while it is important to emphasize the physical interpretation, not just the mathematical relationships, in data-driven approaches. This is why knowledge-guided ML is now highly recommended. Specifically, in the application of ML for explaining ozone formation, emphasis should be placed on feature selection (i.e., variables representing potential drivers) and the interpretation of results.*

*1. In the Methods section, the meteorological normalization method is applied to decouple the impact of meteorology from emission-driven changes in ozone and source-specific VOCs. ML-based meteorological normalization is essentially an adjustment method that aims to correct meteorologically induced variations in air quality time series. Similar statistical approaches have been used since the 1980s in the USA to estimate emission-driven trends of ozone. It is important to clarify that this technique does not "remove" meteorology from observational data but rather reduces its impact through specific techniques. We cannot have air pollution without meteorology. The term "REMOVE" is used throughout the text, it would be prudent to use quotation marks around "REMOVE" to avoid misunderstanding.*

**Response:**

We apologize for our imprecise wording and appreciate the reviewer's comments and suggestions. The machine learning-based meteorological normalization method indeed did not remove the influence of meteorological factors. We have taken your advice and placed "remove" in quotes in the manuscript.

*2. A key question here is the physical meaning of meteorologically normalized ozone. The level of normalized air pollutants depends on how normalization is applied according to the research purpose. Section 2.2 focuses heavily on random forest modeling but lacks sufficient detail about the rationale for feature selection and the meteorological normalization processes, making it difficult for readers to fully understand the implications of the results. The authors discuss the relative importance of dispersion +/ transport and chemistry in driving air pollutants, assuming these atmospheric processes are well represented by variables like wind and air mass clusters. This assumption needs clarification to build confidence in the model results—specifically, what features are proxies for specific atmospheric processes?*

**Response:**

We appreciate the reviewer's comments. We have made the following revisions to clarify which features serve as proxies for specific atmospheric processes:

In the RF model, the air pollutants were the response variables, while the explanatory variables included time variables representing source emissions and meteorological variables representing physical and chemical processes. Time variables such as day of year, month, weekday and hour are used to indicate the seasonal, weekly, and daily cycles of emission intensity (Dai et al., 2023; Vu et al., 2019). Temperature was a key factor influencing the rate of chemical reactions, with higher temperatures typically promoting the photochemical reactions that generate $O_3$. UVB served as the driving force for the photochemical reactions, directly impacting $O_3$ formation. Additionally, humidity played an important role in the chemical processes involved in $O_3$ formation. Therefore, T, RH, and UVB were identified as the key features associated with atmospheric photochemical reactions. WS influences the dispersion of atmospheric pollutants. At high wind speeds, air pollutants tended to be dispersed, while low wind speeds resulted in local pollutant accumulation, leading to increased concentrations. WD determined the dispersion path of atmospheric pollutants. BLH was a critical factor affecting the vertical dispersion of pollutants. A higher boundary layer allowed pollutants to disperse more effectively into the upper atmosphere, reducing surface concentrations, whereas a lower boundary layer resulted in pollutant accumulation near the ground. Thus, WS, WD, and BLH were regarded as the features of atmospheric physical dispersion on a local scale. Cluster can serve as a feature of transport from remote regions.

*3. In Section 2.4, the authors state "In this study, the observed and meteorological normalized VOCs concentrations were fed into US EPA PMF v5.0 to identify and quantify major emission sources of VOCs." This approach is interesting for PMF modeling, particularly in examining changes in source contributions after meteorological normalization to understand the impact of dispersion (should be the overall impact of meteorology) on VOC sources (a good point to address). However, since PMF is a bilinear model requiring additive input variables, questions arise: are these normalized VOCs still additive? How is the total VOC for normalized concentrations calculated? Is the normalized VOC comparable to the observed VOC? An alternative approach to achieve the same goal might be to meteorologically normalize the PMF-resolved source-specific VOCs (i.e., run PMF with observed VOCs first, then normalize each source-specific VOC). This work may be of the authors interest: https://doi.org/10.1029/2023JD038696.*

**Response:**

We appreciate the reviewer's comments. The reviewer raised a very critical point: the mathematical requirement of the PMF model was that the total concentration was a linear combination of contributions from individual sources. Random forest was a nonlinear machine learning algorithm. We applied the random forest model for meteorological normalization to individual VOC species and total VOCs, and found that the sum of the meteorologically normalized VOC species remained linearly correlated with the total VOCs (Fig. S4). Therefore, we believed that the nonlinear processing did not significantly alter the overall structure of the total VOCs concentrations, the results of the PMF model remained reasonable under the conditions. Additionally, we appreciate the reviewer's suggestion regarding the method for meteorological normalization of source-specific VOCs in PMF analysis. We have carefully read and cited this literature, and we believed that this method directly satisfies the additivity principle of PMF, making it an excellent approach. We will also consider comparing the

results of these two methods in the future. We have added the following explanation in Section 2.4 of the Methods:

RF model for meteorological normalization was a nonlinear machine learning algorithm. To satisfy the fundamental mathematical requirement of the PMF model, which stated that the total concentration was a linear combination of contributions from individual sources, the RF model was applied for meteorological normalization of individual VOC species and total VOCs in this study. This ensured that the sum of the meteorologically normalized VOC species remained linearly correlated with total VOCs (Fig. S4), indicating that the nonlinear processing did not significantly alter the overall structure of total VOC concentrations. With this approach, the results obtained by inputting the meteorologically normalized data into the PMF model were reasonable.

[Figure]

**Figure S4: Time series and correlation of the sum of normalized VOC species and normalized total VOCs.**

*4. In Figure 2, all features are ranked with positive values, which describe the magnitude of their impacts without considering the sign of those impacts. However, dispersion can have both positive (enhancing concentration during poor dilution) and negative (reducing pollutant levels)*

*effects. Additionally, using pie charts to illustrate the roles of dispersion and chemistry is problematic because chemistry is not independent of dispersion and transport. Can the authors elaborate more about this?*

**Response:**

We appreciated the reviewer's comments. In this study, we used feature importance to reflect the overall significance of explanatory variables in the RF model. Although these features may have positive or negative effects on $O_3$ concentrations under different environmental conditions, their importance values in the global model quantified the proportion they occupied, reflecting their contribution to the overall prediction performance. While feature importance did not directly reveal whether the influence was positive or negative, it highlighted the critical factors in the model's predictions (Feng et al., 2019; Liu et al., 2022b; Ye et al., 2022; Yang et al., 2023). We appreciated the reviewer for raising this excellent point, and in future research, we planned to employ SHAP analysis to investigate the positive and negative impacts of individual features. We have added the following explanation about feature importance in the Method:

Feature importance was used to reflect the overall significance of explanatory variables in the RF model. The importance was typically represented as an array, where each value corresponded to the importance score of a specific feature. These scores usually range from 0 to 1. The higher importance score indicated that the feature had a stronger predictive capability for the response variable.

Additionally, we were grateful for the reviewer's comment that "chemistry is not independent of dispersion and transport." The pie chart was intended to simplify the display of the relative contributions of emissions, chemical reactions, local dispersion, and long-distance transport. In the Methods section, we added explanations of the atmospheric processes represented by each meteorological parameter:

Temperature was a key factor influencing the rate of chemical reactions, with higher temperatures typically promoting the photochemical reactions that generate $O_3$. UVB served as the driving force for the photochemical reactions, directly impacting $O_3$ formation. Additionally, humidity played an important role in the chemical processes involved in $O_3$ formation. Therefore, T, RH, and UVB were identified as the key features associated with atmospheric photochemical reactions. WS influences the dispersion of atmospheric pollutants. At high wind speeds, air pollutants tended to be dispersed, while low wind speeds resulted in local pollutant accumulation, leading to increased concentrations. WD determined the dispersion path of atmospheric pollutants. BLH was a critical factor affecting the vertical dispersion of pollutants. A higher boundary layer allowed pollutants to disperse more effectively into the upper atmosphere, reducing surface concentrations, whereas a lower boundary layer resulted in pollutant accumulation near the ground. Thus, WS, WD, and BLH were regarded as the features of atmospheric physical dispersion on a local scale. Cluster can serve as a feature of transport from remote regions.

Under this assumption, we considered the interactions between dispersion, transport, and chemistry to be insignificant, allowing us to treat the physical and chemical processes independently. Thank you once again for helping us enhance the scientific accuracy of the paper.

*5. In the Results & Discussion section, the authors demonstrate model performance using only the squared correlation coefficients. It is recommended to also include root mean squared errors, as this is an important metric for describing the accuracy of model predictions.*

**Response:**

We appreciate the reviewer's comments. We have removed the correlation coefficients (r²) from the Results and Discussion section of the manuscript and included r², along with other performance metrics such as root-mean-square error (RMSE), FAC2 (fraction of predictions with a factor of 2), mean bias (MB), mean gross error (MGE), normalized mean bias (NMB), normalized mean gross error (NMGE), coefficient of efficiency (COE), and index of agreement (IOA), in the Supplement.

**Table S2. RF model performance for testing data set .**

| Pollutants | $r^2$ | RMSE | FAC2 | MB | MGE | NMB | NMGE | COE | IOA |
|:---:|:---:|:---:|:---:|:---:|:---:|:---:|:---:|:---:|:---:|
| $O_3$ | 0.88 | 17.33 | 0.80 | -0.34 | 12.70 | -0.01 | 0.22 | 0.68 | 0.84 |
| $NO_2$ | 0.83 | 9.43 | 0.97 | 0.11 | 6.85 | 0.00 | 0.18 | 0.62 | 0.81 |
| NMHCs | 0.76 | 6.41 | 0.99 | -0.11 | 4.60 | 0.00 | 0.20 | 0.54 | 0.77 |

*In summary, I strongly recommend that the authors add more details about feature selection, the adopted meteorological normalization process, and the physical meaning of the normalized air pollutants. One of the existing literature has discussed and reviewed various meteorological normalization strategies based on ML modeling, which may be helpful for this work (https://doi.org/10.1007/s11430-022-1128-1).*

**Response:**

We appreciate the reviewer's comments. We have added the details regarding explanatory variables, model development, model evaluation, and meteorological normalization as follow:

The descriptions of explanatory variables in the response under "General comment 2."

There are approximately 32,856 valid data with a time resolution of 1 hour. The RF model was trained using a forest of 1,000 trees. Training datasets of the RF model was conducted on 80% of the original datasets, and the remaining 20% was selected as testing datasets. Correlation coefficients (r²), root-mean-square error (RMSE), FAC2 (fraction of predictions with a factor of 2), mean bias (MB), mean gross error (MGE), normalized mean bias (NMB), normalized mean gross error (NMGE), coefficient of efficiency (COE), and index of agreement (IOA) were used to evaluate model performance (Table S2). Based on previous related research, these statistical measures indicated that the model performed well (Emery et al., 2017; Henneman et al., 2017; Vu et al., 2019).

The process of meteorological normalization involved replacing the original meteorological variables with those randomly resampled from the observation dataset, and using the established RF model to predict atmospheric pollutant concentrations under different meteorological conditions. The resampling of meteorological variables was conducted over the two-week period before and after the selected date, with the resampled hours remaining constant. This approach effectively preserved the seasonal and diurnal variations in the response variables (Vu et al., 2019). The resampling and prediction process were repeated 1000 times to generate 1000 predicted pollutants concentrations. The average values were taken as the final meteorologically normalized concentrations. In the meteorological normalization process of $O_3$ concentration, meteorological variables such as WS, WD, BLH, and cluster, which signify dispersion and transport, were randomly sampled. In the case of $O_3$ precursors, namely $NO_2$ and NMHCs, resampling was exclusively applied to WS, WD and BLH. $NO_2$ and NMHCs have short atmospheric lifetimes, making them less susceptible to the influence of regional transport over large scales (Wang et al., 2023).

Finanlly, We thank the reviewer for providing us with this excellent work, from which we have learned a lot. We have drawn on the analytical methods and statements, and have cited it in our study.

*Minor Comments*

*1. Line 381: Clarify what is meant by "After normalizing the effect of dispersion."*
*Meteorological normalization is not limited to normalizing the effect of dispersion.*

**Response:**

We appreciate the reviewer's comments. For VOCs, we performed resampling on three meteorological variables: wind speed (WS), wind direction (WD), and boundary layer height (BLH), which we considered mainly to indicate dispersion effects. However, this statement may lead to ambiguity, so we have revised it to the following statement:

"After smoothing out the effect of dispersion"

*2. Figure 8: Regarding the pies for normalized source contributions, are these contributions additive? What is the physical meaning of the sum of normalized source contributions?*

**Response:**

We appreciate the reviewer's comments. The normalized source contributions were still additive as the response in General Comment 3. The physical meaning of the sum of normalized source contributions was the relative contribution of all sources to the total VOC concentration after meteorological normalization. We have added the following description in the manuscript:

Fig. 8 showed the proportion of VOCs sources before and after meteorological normalization during the non-pollution periods and pollution periods. The pies for normalized source contributions illustrated the relative contribution of each source to the total VOC concentration after "removing" the effects of dispersion.

Dai, Q., Dai, T., Hou, L., Li, L., Bi, X., Zhang, Y., and Feng, Y.: Quantifying the impacts of emissions and meteorology on the interannual variations of air pollutants in major Chinese cities from 2015 to 2021, Science China Earth Sciences, 66, 1725-1737, 10.1007/s11430-022-1128-1, 2023.

Emery, C., Liu, Z., Russell, A. G., Odman, M. T., Yarwood, G., and Kumar, N.: Recommendations on statistics and benchmarks to assess photochemical model performance, J Air Waste Manag Assoc, 67, 582-598, 10.1080/10962247.2016.1265027, 2017.

Feng, R., Zheng, H.-j., Zhang, A.-r., Huang, C., Gao, H., and Ma, Y.-c.: Unveiling tropospheric ozone by the traditional atmospheric model and machine learning, and their comparison: A case study in hangzhou, China, Environmental Pollution, 252, 366-378, 10.1016/j.envpol.2019.05.101, 2019.

Henneman, L. R. F., Liu, C., Hu, Y., Mulholland, J. A., and Russell, A. G.: Air quality modeling for accountability research: Operational, dynamic, and diagnostic evaluation, Atmospheric Environment, 166, 551-565, 10.1016/j.atmosenv.2017.07.049, 2017.

Liu, B., Wang, Y., Meng, H., Dai, Q., Diao, L., Wu, J., Shi, L., Wang, J., Zhang, Y., and Feng, Y.: Dramatic changes in atmospheric pollution source contributions for a coastal megacity in northern China from 2011 to 2020, Atmospheric Chemistry and Physics, 22, 8597-8615, 10.5194/acp-22-8597-2022, 2022a.

Liu, H., Yue, F., and Xie, Z.: Quantify the role of anthropogenic emission and meteorology on air pollution using machine learning approach: A case study of $PM_{2.5}$ during the COVID-19 outbreak in Hubei Province, China, Environmental Pollution, 300, 10.1016/j.envpol.2022.118932, 2022b.

Liu, Y., Geng, G., Cheng, J., Liu, Y., Xiao, Q., Liu, L., Shi, Q., Tong, D., He, K., and Zhang, Q.: Drivers of Increasing Ozone during the Two Phases of Clean Air Actions in China 2013–2020, Environmental Science & Technology, 57, 8954-8964, 10.1021/acs.est.3c00054, 2023.

Vu, T. V., Shi, Z., Cheng, J., Zhang, Q., He, K., Wang, S., and Harrison, R. M.: Assessing the impact of clean air action on air quality trends in Beijing using a machine learning technique, Atmospheric Chemistry and Physics, 19, 11303-11314, 10.5194/acp-19-11303-2019, 2019.

Wang, Y., Jiang, S., Huang, L., Lu, G., Kasemsan, M., Yaluk, E. A., Liu, H., Liao, J., Bian, J., Zhang, K., Chen, H., and Li, L.: Differences between VOCs and NOx transport contributions, their impacts on O3, and implications for $O_3$ pollution mitigation based on CMAQ simulation over the Yangtze River Delta, China, Science of the Total Environment, 872, 10.1016/j.scitotenv.2023.162118, 2023.

Yang, C., Dong, H., Chen, Y., Wang, Y., Fan, X., Tham, Y. J., Chen, G., Xu, L., Lin, Z., Li, M., Hong, Y., and Chen, J.: Machine Learning Reveals the Parameters Affecting the Gaseous Sulfuric Acid Distribution in a Coastal City: Model Construction and Interpretation, Environmental Science & Technology Letters, 10, 1045-1051, 10.1021/acs.estlett.3c00170, 2023.

Yang, J., Zeren, Y., Guo, H., Wang, Y., Lyu, X., Zhou, B., Gao, H., Yao, D., Wang, Z., Zhao, S., Li, J., and Zhang, G.: Wintertime ozone surges: The critical role of alkene ozonolysis, Environmental Science and Ecotechnology, 22, 10.1016/j.ese.2024.100477, 2024.

Ye, X., Wang, X., and Zhang, L.: Diagnosing the Model Bias in Simulating Daily Surface Ozone Variability Using a Machine Learning Method: The Effects of Dry Deposition and Cloud Optical Depth, Environmental Science & Technology, 56, 16665-16675, 10.1021/acs.est.2c05712, 2022.